# Insights into Four *NAC* Transcription Factors Involved in Grain Development and in Response to Moderate Heat in the *Triticeae* Tribe

**DOI:** 10.3390/ijms231911672

**Published:** 2022-10-02

**Authors:** Claire Guérin, Céline Dupuits, Said Mouzeyar, Jane Roche

**Affiliations:** UMR 1095 Génétique, Diversité et Ecophysiologie des Céréales, Université Clermont Auvergne, INRAe, 63170 Aubière, France

**Keywords:** grain, heat stress, NAC, transcription factor, *Triticeae*, wheat

## Abstract

NAC (NAM (no apical meristem)–ATAF (Arabidopsis transcription activation factor)–CUC (cup-shaped cotyledons)) are among the largest transcription factor families in plants, involved in a plethora of physiological mechanisms. This study focused on four *NAC* genes previously identified in bread wheat as specifically grain-expressed which could be considered as candidate genes for yield improvement under climate changes. Using in silico analyses, the *Triticum aestivum* “Grain-NAC” (*TaGNAC*) orthologs in 14 cereal species were identified. A conserved protein motif was identified only in *Triticeae*. The expression of *TaGNAC* and einkorn *TmGNAC* was studied in response to moderate heat stress during grain development and showed a similar expression pattern that is accelerated during cell division stages under heat stress. A conserved structure was found in the promoter of the *Triticeae* *GNAC* orthologs, which is absent in the other *Poaceae* species. A specific model of promoter structure in *Triticeae* was proposed, based on the presence of key cis-elements involved in the regulation of seed development, hormonal regulation and response to biotic and abiotic stresses. In conclusion, *GNAC* genes could play a central role in the regulation of grain development in the *Triticeae* tribe, particularly in the accumulation of storage proteins, as well as in response to heat stress and could be used as candidate genes for breeding.

## 1. Introduction

NAC (NAM (no apical meristem)–ATAF (Arabidopsis transcription activation factor)–CUC (cup-shaped cotyledons)) are one of the largest families of transcription factors in the plant kingdom [1]. This family have been described for numerous plant species, for example, Arabidopsis (117 members) [2], tobacco (154 members) [3], soybean (180 members) [4], *Fragaria × ananassa* fruits (112 members) [5], tomato (93 members) [6] and *Cleistogenes songorica* desert grass (162 members) [7]. During the past few years, *NAC* genes were particularly studied in cereals, as key regulators of plant development and potential yield improver. In this sense, NAC were listed in a plethora of cereals species, such as rice (151 members) [2], maize (157 members) [8], *Brachypodium distachyon* (118 members) [9], durum wheat (168 members) [10], barley (167 members) [11], and more recently bread wheat (488 members) [12].

NAC proteins are characterized by their highly conserved NAC domain located in the N-terminal region of the protein. This DNA-binding domain is approximately 150 amino acids (aa) in length and is subdivided into five subdomains named A to E [13]. The NAC domain is also responsible for the homo- or heterodimerization of the NAC proteins [14]. A transcriptional regulatory region is present in the C-terminal region of the protein. This poorly conserved domain acts as a transcriptional repressor [15] or activator [16] of target genes and often contains simple amino acid repeats and regions rich in serine and threonine, proline and glutamine, or acidic residues [17].

Increasing attention has been given to *NAC* genes because of their central role in many signalling pathways in response to diverse plant growth conditions, particularly due to their role in plant organ development [17]. During the vegetative stage, the NAC family regulates, among other physiological processes, root development [16], flowering [18], leaf senescence [4], secondary cell wall biosynthesis [19] and cell division [20]. At the reproductive stage, *NAC* genes are involved in embryogenesis [21] and seed development [22], and have been also described as actors involved in the elaboration of grain yield and quality. As an example, a reduced expression of *OsNAP* in rice and *TaNAC*-S in wheat delay leaf senescence, leading to a higher grain yield [23].

Moreover, NAC proteins are also involved in the response to abiotic stresses, including drought, salt, and temperature [24]. In response to high temperature and water stress, for example, several NAC are at the centre of regulation pathways by regulating other transcription factors which are specifically responding to abiotic responses. For example, JUB1/ANAC042 (JUngBrunnen1) has been demonstrated as a positive regulator of tolerance to heat, salt, and water stress in Arabidopsis [25,26] and tomato [27]. AtNAC019 is also a transcription factor involved in temperature signal mediation in Arabidopsis by binding to the promoters of *HSFA1b*, *HSFA6b*, *HSFA7a*, and *HSFC1* genes [28]. In cereals, an NAC transcription factor has been proven to activate the expression of temperature-related genes (*DREB2A*, *DREB2B*, *AtCYP181*, *RD17*, *HSP26.5*, *LEA*, *AtGolS1*, *HSP70*, *AtHsfA3* and *RD29A*) in wheat (*TaNAC2L* [29]) or in rice (*OsNTL3*/*OsNTL4* [30,31,32]; *SNAC3* [33]) and also in maize (7 *ZmNTLs* [34]). Morishita et al. (2009) showed that in response to high light and temperature stress, overexpression of the *ANAC078* gene induces activation of the expression of more than 166 genes in Arabidopsis [35].

As one of the largest families, *NAC* genes have been demonstrated to have a tissue-specific expression. *TaNAC2a*, *TaNAC2D*, *TaNAC4a*, *TaNAC6*, *TaNAC7*, *TaNAC13* and *TaNTL5* are induced by water stress, high salinity and/or low temperature [36,37]. These genes are localized in the nucleus or bound to the membrane and five of them possess an organ-specific expression. In wheat, Zhao et al. (2015) found an increase in nitrogen concentration in grains derived from lines over-expressing the TaNAC-S protein, which is an NAC factor expressed primarily in leaves [38]. It would lead to a delayed post-anthesis foliar senescence (stay-green phenotype) and better nutrient remobilization during this senescence while maintaining the same yield as the control lines. Similarly, in barley, an NAC transcription factor subclade (NAC-d-9) has been described as exclusively expressed in the grains, and may be involved in programmed cell death of the endosperm occurring during grain maturation [11]. In rice, Mathew et al. (2016) also showed that *ONAC20* and *ONAC26* are expressed specifically during rice seed development at extremely high levels and are associated with seed size [39]. They are suggested to be exploited as potential targets for crop improvement.

Here, we studied four grain-specific genes previously identified in bread wheat [12]. We showed that, in bread wheat, the four *NAC* genes are preferentially expressed during the grain cell division stage, and that their expression is altered in response to moderate heat stress applied during grain development. In addition, their protein sequences were studied and compared with the sequences of their orthologs in 13 other cereal species. The evolutionary pattern of those orthologs among species was studied. Globally, the structure of those genes and their promoters have been studied in the *Triticeae* tribe and showed a conserved structure that supports the hypothesis of a central role of those genes in seed development in response to heat stress.

## 2. Results

### 2.1. Identification of the Four NAC Genes Grain-Expressed (GNAC) in 15 Cereal Species

Based on our previous work on the NAC family [12], we focused on four genes belonging to the NAC family in bread wheat that are specifically expressed in the grain during the reproductive stages. As suggested by Murozuka et al. (2018) [11] concerning their orthologs in barley, those genes were named *Grain-NACs*: *GNAC1*, *GNAC2*, *GNAC3* and *GNAC4*. The two letters in front of the gene indicate the species they derived. For example, *TaGNAC1* was used for *GNAC1* from *Triticum aestivum* (bread wheat).

Using the IWGS RefSeq v2.1, the latest annotation of the bread wheat genome, we retrieved the genomic, coding and protein sequences of *TaGNAC1*, *TaGNAC2*, *TaGNAC3* and *TaGNAC4* genes in the Chinese Spring bread wheat variety (Appendix A). The genomic sequences of *TaGNACs* are between 1085 and 1710 base pair (bp) length. They all have a single intron, which is not the most common structure among NACs (only 19% of NACs) as the majority have two introns (44% of NACs) [12]. Intron size is conserved between homeologs of *TaGNAC2* (110 bp for all three homeologs), *TaGNAC3* (110 bp), and *TaGNAC4* (113 bp), but there is a length variation in the intron of the three *TaGNAC1* homeologs A, B, and D, which are 565, 572, and 573 bp length, respectively (Appendix A). The coding DNA sequence (CDS) length of the four *TaGNAC* is variable. *TaGNAC1-7A* is the longest of the set (1143 bp) while *TaGNAC3-5A* is the shortest (984 bp). In addition, the length of the cDNAs varies between homeologs for all four genes studied. For example, the three *TaGNAC1* homeologs have a variable cDNA length (1143 bp, 1068 bp and 1095 bp for homeologs A, B and D, respectively). This variability is explained by the identification of the starting ATG, which is not the same between homeologs and leads to variability in the protein length (381 aa for TaGNAC1-7A, for 346 aa for TaGNAC1-7B and 365 aa for TaGNAC1-7D). Similarly, *TaGNAC4-7A* and *TaGNAC4-7D* have a length of 1050 bp while *TaGNAC4-7B* is 972 bp length. This 78 bp truncation is, in that case, due to the appearance of a premature stop codon on the *TaGNAC4-7B* sequence, truncating the end of the protein by 26 aa (324 aa vs. 350 aa for the A and D homeologs). The protein sequences of the four TaGNACs all have the signature domain of the NAC family (PF02365). The four TaGNAC and the four TmGNAC protein sequences harbor a probable non-cytoplasmic location (more than 80% of probability) without any signal peptide nor transmembrane helix for all the sequences. This result is consistent with the observations made by Guérin et al. (2019) on an older version of the RefSeq, showing that only 18 *TaNAC* sequences carry a transmembrane helix, and they belong mainly to the NAC-b subfamily [12].

The same approach was used to retrieve sequences from einkorn. The four *TmGNAC* sequences show similar lengths to their A orthologs in bread wheat, except for *TmGNAC3*, truncated by 279 bp (93 aa) compared to *TaGNAC3-5A* (Appendix A). This difference in length is explained by a gap located in the C-terminal region of the *TmGNAC3* sequence and which corresponds to the portion between bp 633 and 911 (aa 211 and 304) of the *TaGNAC3* sequence. The NCBI Global Align tool was used to calculate the percentage of identity between the full-length sequences of each *TaGNAC* and their *TmGNAC* orthologs (Appendix A). The coding sequences of *TmGNAC1* and *TmGNAC4* show very high homology with their orthologs *TaGNAC1* and *TaGNAC4* (95% and 97% identity, respectively). On the contrary, *TmGNAC2* and *TmGNAC3* show only 67% identity with their respective orthologs *TaGNAC2* and *TaGNAC3*. For *GNAC3*, this is explained by the presence of the above-mentioned gap. For *GNAC2*, this is explained by mismatches distributed along the entire length of the alignment. Like their TaGNAC homeologs, all TmGNACs have the NAC domain in their protein sequences. Similarly, they do not have a transmembrane domain or a peptide signal.

To study the sequence conservation of those genes among cereals, three homeologs of the four TaGNAC proteins were blasted against the whole proteome of 13 cereal species and the best hit was selected for each result (Appendix A). We manually added the protein sequences of the four *T. monococcum* GNAC (TmGNAC1, TmGNAC2, TmGNAC3 and TmGNAC4) previously sequenced to this pool of orthologous sequences. Furthermore, the presence of the NAC domain (IPR003441, PS51005 and PF02365) was validated for all the sequences; along with a biological process of regulation of transcription, DNA-templated (GO:0006355) and a molecular function of DNA-binding (GO:0003677). Those results are consistent with the NAC protein role as transcription factors. No peptide signal nor transmembrane domain was identified on those orthologous sequences.

A high conservation was observed in GNAC1, GNAC2, GNAC3 and GNAC4 sequences among the *Triticeae* group (*Triticum turgidum*, *T. aestivum*, *T. spelta*, *T. dicoccoides*, *T. monococcum*, *T. urartu*, *Aegilops tauschii* and *Hordeum vulgare*) (Figure 1). Indeed, except for the TmGNAC2, which was classified in the “GNAC4 clade”, all the *Triticeae* orthologs of GNAC1, GNAC2, GNAC3 and GNAC4 are clustered in separate clades. This high level of sequence conservation suggests putative conservation of the function of those genes between the *Triticeae* species. The presence of the *TmGNAC2* gene in the “GNAC4 clade” can be explained by the fact that the *GNAC2* and *GNAC4* genes present a strong sequence similarity. It also has to be noticed that the *HORVU7Hr1G031260.1* gene has been identified as an ortholog of *TaGNAC2* and *TaGNAC3*, which explains why it is named *HvGNAC2* and *HvGNAC3* on the phylogenetic tree. However, it is probably closer to *TaGNAC2* than to *TaGNAC3* since it is attached to the “GNAC2” clade on the phylogenetic tree. We can therefore hypothesize that the *TaGNAC3* gene from bread wheat does not have a direct ortholog in *Hordeum vulgare*. However, for the seven non-*Triticeae* species, the orthologous sequences of the *TaGNAC* genes seem to be less organized (they are orthologous to several *TaNAC* genes). For example, in the case of two species (*B. distachyon* and *S. italica*), there is only one orthologous sequence corresponding to all the homeologs of the four *TaGNACs* genes: the sequence *KQK02751* in *B. distachyon*, and the sequence *KQL24250* in *S. italica*.

### 2.2. Structure and Functional Domains of GNAC Proteins

To investigate the conservation of the protein structure in the NAC preferentially expressed in grain, we modelled the tertiary structure of the TaGNAC and TmGNAC as an example (Figure 2). This result shows that the structure of the studied proteins is conserved, with four main beta-sheets and two alpha-helix in a central position of each protein.

After comparing the set of 488 TaNACs from the RefSeqv1.0 database [12], only 454 sequences were retrieved from the new protein database (RefSeq v2.1). The presence of conserved motifs was investigated among these protein sequences. Among the 20 motifs identified, a 68 aa motif was present in the C-terminal part of 17 sequences belonging to a clade of the phylogenetic tree consisting of 21 sequences in total (Figure 3) corresponding to the NAC-d subfamily, with an e-value of 1.9 × 10^−182^. Guérin et al. (2019) identified 19 *TaNAC* genes preferentially expressed in grain [12]. Sixteen of these 19 sequences are contained in the same clade. Of these, 14 sequences carry the 68 aa-motif.

The 21 genes that constitute the clade are structured into several homeology groups (triad, dyad and singleton). This clade carries the three homeologs of *TaGNAC1* and *TaGNAC4*, structured in triads. These two triads have a strong sequence homology with two homeology groups, respectively, (*TraesCS7A03G1388800*.1 and *TraesCS7D03G1305900.1* for *TaGNAC1*; and *TraesCS7B03G0248800.1*, *TraesCS7D03G0426000.1* and *TraesCS7A03G0440800.1* for *TaGNAC4*). From an evolutionary point of view, these homeologous groups may be the result of a phenomenon of post-hexaploidization duplication. *TaGNAC3* has only the homeolog A, so it is represented by only one sequence in this clade. This sequence has strong homology with the *TraesCS3A03G0172000.1*, *TraesCS3B03G0216600.1* and *TraesCS3D03G0154500.1* homeology group. The *TaGNAC2* homeologous group is less clearly structured in this clade. Indeed, it appears to form a group of five sequences, four of which are carried by chromosome 7 and one sequence is anchored on chromosome 3. The 21 genes anchored in this clade have a similar two exons–one intron structure. However, this configuration represents only 19% of the members of the TaNAC family, because the majority (44%) of the *NAC* sequences have three exons and two introns.

It is known that a transcriptional activity-regulating domain is located in the C-terminal part of the NAC proteins. This domain is less conserved than the NAC domain and, consequently, weakly described [17]. The identification of a motif conserved in the N-terminal part of the TaNAC sequences expressed preferentially in the grain and gathered within a specific clade of the phylogenetic tree supports the potential existence of this domain of activity regulation within these proteins. It is also suggested that, potentially, genes carrying this motif have a similar function within bread wheat [11]. To study the conservation of this motif within the orthologous GNAC proteins of bread wheat (and thus, potentially, the conservation of the regulatory function of the transcriptional activity of the *GNAC* genes), we have conducted a FIMO analysis. This analysis identified the presence of the motif in 47 sequences among the 64 orthologous sequences in 15 species (*T. aestivum*, *T. monococcum* and the 13 other species) (Appendix A). This motif is present in GNAC orthologs of only all *Triticeae* species, except for TmGNAC3. The conservation of this motif within the GNACs of *Triticeae* suggests strong conservation of the C-terminal sequence of these proteins. It could correspond to the regulation domain (activator/repressor) of the transcriptional activity of proteins, which would reflect conservation of function in *Triticeae*. The pattern is not found in the TmGNAC3 protein because the truncated part in TmGNAC3, compared to TaGNAC3, is precisely the part where the motif was identified.

### 2.3. Expression of Orthologs of the Four TaGNAC Genes in Einkorn Grain (TmGNAC) and TaGNAC Homeologs during Development

In a previous work, we showed that *TaGNAC1*, *TaGNAC2*, *TaGNAC3* and *TaGNAC4* present a preferential expression in the bread wheat grain [12]. Among a large panel of cereals, their sequences are particularly conserved within the *Triticeae* group. Thus, we investigated the expression profile in *T. monococcum* (einkorn) to determine whether a similar expression pattern may be found in another *Triticeae*. To answer this question, we followed the expression of the four *TmGNACs* (diploid carrier of the bread wheat A genome) and the four *TaGNACs* carried by the bread wheat A copy, during grain development.

First, the expression of the four *TmGNAC* genes in einkorn was monitored on eight developmental stages under control conditions (19 °C). The analyzed stages have been selected at the early phase of cell divisions, and the beginning of the filling phase. The last time point (700 °Cd (degree Celsius day after anthesis)) corresponds to harvest at seed maturation. Globally, the expression level of *TmGNACs* is low during the early phases of grain development and increases around 120 °Cd to reach a maximum phase at 300 °Cd (Figure 4). This profile is similar for the four genes studied, however, the maximum expression value is variable between genes. Specifically, *TmGNAC2* is the gene that reaches the highest expression level (at 300 °Cd = 16.6 in the control condition) while *TmGNAC3* is the lowest expressed gene among the four genes studied (at 300 °Cd = 0.0463 in the control condition). *TmGNAC1* and *TmGNAC4* show a similar expression intensity = 10.4 and 8.0 at 300 °Cd, in the control conditions. The last time point (700 °Cd after anthesis) corresponding to the end of the maturation phase reveals a level of expression of the four *GNAC* similar to the level of expression before 300 °Cd, showing that the expression of the four *GNAC* genes does not increase exponentially during the growth of the grain but decreases during the late phases of its development.

In parallel, using RT-qPCR, we also monitored the expression level of each *TaGNAC* homeolog in bread wheat grain during six early stages of development (Figure 5). The three homeologs are differentially expressed under control conditions depending on the *TaGNAC*. Regarding *TaGNAC1*, *TaGNAC2* and *TaGNAC4*, homeolog A is the least expressed. In *TaGNAC1*, the B and D copies are expressed at equivalent levels (11.11 and 11.40). The most highly expressed homeolog in *TaGNAC2* is the B copy, while in *TaGNAC4* it is the D copy. In the domestication history, crossing *Triticum monococcum* (AA) with a closely related species of *Aegilops speltoides* (BB) gave rise to durum wheat (AABB), itself an ancestor of hexaploid bread wheat (AABBDD). The einkorn species, therefore, carries the AA genome from the same ancestral genome as the AA subgenome of bread wheat. We, therefore, focused here on the A homeologs of each of the four *GNACs*. The observed profiles are similar for the four *TaGNAC* genes: a first minimal phase is observed from 0 °Cd to 120 °Cd, and then the relative expression level of *TaGNACs* increases progressively to reach its maximum value at 300 °Cd, which is the last stage measured. However, while the expression profile under control conditions is similar between *TaGNAC1-7A*, *TaGNAC2-7A*, *TaGNAC3-5A* and *TaGNAC4-7A*, their relative expression level is variable. As observed for einkorn, *TaGNAC3-5A* is very weakly expressed (0.3). However, unlike einkorn, *TaGNAC2-7A* is not the most expressed A-homolog of the four *TaGNACs*, it has a much lower expression than *TaGNAC1-7A* and *TaGNAC4-7A* (respectively, 0.52 vs. 5.26 and 2.21).

### 2.4. Gene Expression Profile of the TmGNAC in Einkorn and TaGNAC A Homeologs in Bread Wheat in Response to Heat Stress

The effect of moderate heat stress was evaluated on the expression of the four *TmGNACs* in the same condition as the experiment carried out on bread wheat (Figure 4). The expression of *TmGNAC1*, *TmGNAC2* and *TmGNAC4* increases later under heat stress conditions compared to the control. Indeed, significantly lower expression is observed at 120 °Cd in response to temperature compared to the control. This delay in expression is quickly compensated in the case of the *TmGNAC1* and *TmGNAC4* genes since a stronger expression is then observed in response to heat stress at 180 °Cd. It is then delayed again at 220 °Cd since the expression in the control condition is significantly stronger than the expression in the stressed condition for *TmGNAC1*, *TmGNAC2* and *TmGNAC3*. Moreover, at 300 °Cd, the expression level is higher in response to heat stress than under control conditions. However, this temperature response disappears afterwards, since the expression level of the four *TmGNACs* is equivalent to or even lower for the 700 °Cd point.

The expression level of the 10 homeologs was monitored during seed development also in response to heat stress. Overall, the expression profile of the four *TaGNAC* homeologs is similar in the control condition and in response to moderate heat stress, i.e., the expression is minimal in the early phases, then increases during the cell division phase of grain development and finally reaches a plateau around 300 °Cd (Figure 5). However, the phase of increased expression seems to be earlier in response to heat stress. Indeed, a statistically higher level of expression has to be noticed in the stressed condition compared to the control condition at 120 °Cd (except for TaGNAC1-7D) and 220 °Cd stages. However, either in the control condition or in response to temperature, the relative expressions between each *TaGNAC* remain similar; *TaGNAC3-5A* shows the lowest expression, followed by *TaGNAC2-7A*; while *TaGNAC1-7B* and *TaGNAC4-7D* possess the highest expression. To estimate more accurately the effect of the stress response, the ratio of the expression under stressed vs. control conditions was performed for each homeolog and each stage in bread wheat (Figure 6). The results indicated that, despite a variable intensity of expression between the genes studied, there was a maximal difference at 120 °Cd after anthesis between control and stressed conditions for all the homeologs studied, suggesting a key stage of response to heat stress at that particular stage. Indeed, except for *TaGNAC1-7D*, which shows an almost non-existent response to temperature (ratio stress/ctrl = 1.05), the three *TaGNAC4* homeologs present the weakest response to stress (3.91, 4.54, 4.95, respectively, for *TaGNAC4-7B*, *TaGNAC4-7D* and *TaGNAC4-7A*). In contrast, *TaGNAC2-7D* shows the highest ratio, with a 22.06-fold higher expression level in response to stress, compared to the control condition. Similarly, *TaGNAC3-5A*, which shows the lowest relative expression level among the four genes (=10 homeologs) studied, is the 2nd gene whose expression is the most increased in response to heat stress (18.27 times more cDNA in the stressed 120 °Cd grain samples than in the controls). Thus, some genes show high expression during development but a weak response to stress (such as *TaGNAC4* homeologs) while others are more weakly expressed during development but respond to stress significantly (such as *TaGNAC3-5A*). Regarding the three *TaGNAC1* homeologs, homeologs 7B and 7D are the most highly expressed at all stages. However, the ratio of stressed vs. controlled expression at 120 °Cd shows that the homeolog A is upregulated under stress, compared to the control condition. For *TaGNAC2*, a similar result was observed: *TaGNAC2-7D* shows a relatively low expression level but the highest stress response amplitude among the three homeologs.

### 2.5. Prediction of the Transcriptional Regulation of the GNAC Expression

The presence of conserved motifs was investigated in the 2000 bp of the promoter region upstream of the *TaGNAC* CDS in bread wheat and in the 13 orthologous species. The results showed a conservation of the promoter sequence of the orthologous *GNACs* in the *Triticeae* tribe (Figure 7). Indeed, 10 motifs were identified exclusively in *Triticeae* promoters. In contrast, none of the *GNAC* promoters from other species were identified as carrying a conserved motif among the 10 identified. This finding supports the hypothesis of a conserved sequence between the orthologous genes *GNAC1*, *GNAC2*, *GNAC3* and *GNAC4* in the *Triticeae*, which is not found in other phylogenetically more distant species. This result observed at the promoter sequence level matches with what has already been observed for the coding sequence of these genes, suggesting a high conservation throughout the *Triticeae* tribe.

Figure 7 reveals the existence of 10 motifs, all anchored on *Triticeae GNAC* sequences exclusively but variably distributed within these sequences. Motifs 1 and 4 are motifs anchored in the 3’ part of the CDS, near the start codon. They are present in almost all the sequences studied (except one and two sequences, respectively) and mostly associated with motif 7, also anchored in the 3’ part of the promoters. It has to be noticed that in some *GNAC2* sequences, motif 4 is not anchored next to motifs 1 and 7 in the normal direction but anchored in the reverse direction and further upstream on the promoter. Motifs 2 and 6 are present on all *GNAC1* promoters, and only *GNAC1*. They are systematically anchored in the vicinity of motifs 1, 4 and 7. Motif 9 is present only in *GNAC2* orthologous sequences. Motifs 3, 5 and 8 are anchored in the centre of the promoter. They are present on almost all *GNAC4* and *GNAC1* orthologous sequences, and only on them. The motif furthest from the 5’ end of the promoter is motif 10. It is found on all *GNAC2* and *GNAC4* sequences, and only on them. *TaGNAC3* and its orthologous sequences do not have a specific motif. Indeed, unlike the other *GNAC* promoters studied, they only carry the most recurrent motifs 1, 4 and 7, which are also found on *GNAC1*, *GNAC2* and *GNAC4*.

In a second step, we identify consensus sequences already described in the literature as transcription factor interaction sites (Appendix A). The 10 identified motifs correspond significantly (e-value < 10) to consensus sequences already described in the literature as playing a role during development and in the response to biotic and abiotic stresses in plants. In motifs 1, 4 and 7 that were found in the promoters of all four *GNACs*, the TOMTOM tool identified consensus sequences involved in the developmental processes (*REM1*, *RVE1*, *PIF4* and *PIF5*), in response to biotic (*AHL20_2* and *WOX13_2*) but also abiotic stresses (*CCA1*, *MYC2*, *PIF4* and *PIF5*). Among these, we find sequences of response to thermal stress (*HSFC1*, *HSFB2A_2*, *HSFC1_2*). Between the two motifs specific to *GNAC1* gene promoters, motif 2 seems to carry the majority of consensus sequences related to the response to abiotic stresses, while motif 6 carries signature sequences for intervention in developmental processes. Motif 9, anchored on the promoters of the *GNAC2* genes only, carries mainly developmental (*WRKY12*, *TCP16*, *WRKY45* and *Dof5.7_2*) and abiotic stress sequences, especially salt and osmotic stress (*DDF1*, *MYB46_2*, *RAP2.3_3*) consensus sequences. The motifs 3, 5 and 8 that we observed on the *GNAC4* and *GNAC1* sequences carry consensus sequences linked to development (*ARR14*, *YAB5*, *AHL12*, *KAN1*, *ATHB51*, *LBD16*, *REM1*, *RVE1*) but some consensus sequences related to biotic stress were found repeated several times on motif 3 (*AHL20* and *ATHB12*), as well as two sequences of response to cold (*DAG2* and *CCA1*). Motif 5 carries two consensus sequences related to hormone regulation (*LBD16* and *ATERF1*). Motif 10, which exists on the *GNAC2* and *GNAC4* promoters, carries developmental sequences (*RVE1*, *YAB5*, *AHL12*) and one biotic stress response sequence (*AHL20*) but no abiotic stress response.

## 3. Discussion

Within the TaNAC family, we found a specific clade that carries genes predominantly expressed in the wheat seed [12]. Among these genes, in this study, four genes were identified as differentially expressed during the seed developmental stages and are strongly impacted by heat stress.

### 3.1. TaGNAC Genes Carry a Recognizable Pattern and Are Highly Conserved through the Triticeae Tribe

The phylogenetic study underlines that several *GNAC* sequences in bread wheat are orthologs of one gene in rice, maize, or *Brachypodium distachyon* for example. This result is consistent with the observation of Murozuka et al. (2018), who unraveled the expansion of a particular clade within the *Triticeae* compared to other *Poaceae* species (rice and maize) [11]. This clade is involved in grain development and, according to the authors, its expansion would correlate with the development of *Triticeae* in a Mediterranean environment that required the establishment of rapid grain maturation mechanisms. Related to the expansion mechanisms of this gene cluster, Murozuka et al. (2018) identified 10 tandem and 8 segmental duplications related to *NAC* genes in barley [11]. Considering that *TaNAC* genes with homologous functions seem to be clustered in the same subfamilies [40], we can speculate that a specific expansion of a subfamily via duplications could be linked to the evolution of plant adaptation mechanisms by multiplying genes linked to specific pathways. This hypothesis may be linked to different duplication events that may have occurred in the *NAC* gene family leading to the functional diversification of the gene family’s members [12]. This is the hypothesis proposed by Murozuka et al. (2018) to explain the appearance of the clade carrying genes involved in grain development [11]. Therefore, if a clade is derived from a duplication event, it makes sense to find a similar genomic structure between the sequences. Thus, Murozuka et al. (2018) identified seven genes (*HORVU4Hr1G089450.1*, *HORVU7Hr1G039700.1*, *HORVU7Hr1G122680.1*, *HORVU3Hr1G014090.1*, *HORVU7Hr1G031260. 1*, *HORVU3Hr1G014100.1* and *HORVU2Hr1G082320.2*) grouped in a clade and showing strong and almost exclusive expression in the developing grain. These authors proposed to call these genes “Grain-NACs”. Six of these seven genes belong to the NAC-d subfamily and, among them, four are orthologs of the *GNAC* genes we studied (*HORVU4Hr1G089450* = *TaGNAC4*; *HORVU7Hr1G031260* = *TaGNAC2*; *HORVU7Hr1G039700* = *TaGNAC4*; *HORVU7Hr1G122680* = *TaGNAC1*). Due to their preferential expression in the grain, we can suggest that these NACs play a role in the formation, differentiation and/or development of the barley and bread wheat grain, and potentially in the establishment of the storage tissues required during the seed development.

We observed the presence of a conserved motif in the C-terminal region of the protein sequences of TaGNAC1, TaGNAC2, TaGNAC3 and TaGNAC4. This motif is present in all 10 existing bread wheat homeologs and only in these GNAC proteins. By comparing the GNACs of bread wheat and their orthologs in other cereals, we found that only the GNACs of species belonging to the *Triticeae* group carry this motif. The presence of a motif can be linked to one or more biological functions [41,42,43]. We can therefore hypothesize that the presence of this motif would be linked to a conserved function of these genes in the *Triticeae* and only in them. The role of this motif has not yet been characterized. In rice, it would not be involved in DNA binding and the potential heterodimerization of GNACs since these roles are performed by the NAC domain [39]. Indeed, they showed the existence of three NAC transcription factors (*ONAC20*, *ONAC23*, and *ONAC26*) expressed specifically during grain development that are associated with rice grain size and grain weight. Their role is thought to be related to grain development with redundant functions. A blastP of their protein sequences on RefSeq v2.1 shows that ONAC20 and ONAC26 are orthologous to GNAC1 (< e-60). In rice, ONAC026 and ONAC020 heterodimerize and this heterodimer is relocalized in the nucleus. This situation could be identical in wheat for the four GNACs studied. The presence of this conserved motif in the C-terminal region of TaGNACs preferentially expressed in the grain is consistent with the results observed by Murozuka et al. (2018) [11]. These authors found that the six “Grain-NACs” in barley carry a 175 aa C-terminal portion conserved at 51.5–64.5% identity, whereas it is 25% maximum when this C-terminal portion is compared with the set of other NACs in barley. Using the same hypothesis, we then could suggest that the Grain-NACs would be the result of recent duplication events (during the divergence of the *Triticeae* inside the *Poaceae* group) that led to the expansion of this specific subgroup and the multiplication of the genes carrying this motif in wheat.

### 3.2. The Four GNACs Harbour a Similar Expression Pattern during Seed Development in Wheat and Einkorn

In a recent global transcriptome analysis of developing wheat grains, Guan et al. (2022) identified 5718 transcription factors during the wheat grain expanding phase, accounting for 5.46% of the total DEGs [44]. Among them, *NAC* genes were the second most representative category (111 genes). In our study, we decided to focus specifically on genes that we have shown to be preferentially expressed in the grain. The maximum expression of the four *TaGNAC* genes in wheat and the four *TmGNAC* genes in einkorn was observed at the 300 °Cd stage of grain development in our conditions. This stage corresponds to the beginning of the wheat grain-filling phase. It suggests that the four *GNAC* gene expressions could be correlated with the accumulation of grain reserves (starch and protein). This hypothesis is supported by the observations made by Plessis et al. (2013) who showed a correlation between the peak of GSP accumulation (grain storage proteins) genes and the maximal peak of *GNAC1* (named *NAC22* in their publication) and *GNAC3* (=*NAC27*) expression in bread wheat [45]. Furthermore, using GWAS analysis, a strong linkage disequilibrium between markers carried by *GNAC1* (=*NAC22*) and *GNAC3* (=*NAC27*) and two markers associated with the glutenin/total nitrogen ratio was noticed. Another in silico study of transcriptional regulatory networks showed that *GNAC3* and *GNAC4* (=*GNAC27* and *GNAC18*, respectively) are strongly linked to the transcription factor network that controls the expression of GSP [46]. Moreover, in maize, Zhang et al. (2019) identified two endosperm-specific *NAC* transcription factors, *ZmNAC128* and *ZmNAC130*, which specifically activate the transcription of the storage protein 16-kDa γ-zein, which are alcohol-soluble prolamins [47]. The equivalent storage proteins in bread wheat would be the gliadins which are also a family of alcohol soluble prolamins. Performing a blast with these two maize *NAC* genes, we found that they are orthologs of *TaGNAC1* and *TaGNAC2*, respectively. This result may suggest that the *TaGNAC* genes may be involved in the regulation of GSP accumulation, and more specifically gliadins, as their orthologs are in maize.

In the *GNAC* promoters, conserved motifs were identified between different species. A study [11] showed the presence of cis-regulatory elements in the promoter regions of “*Grain-NAC*” genes in barley, some of them are orthologs of the four *TaGNAC* genes. They also showed the presence of these cis-elements in *TaGNAC4* and *TaGNAC2* in bread wheat. These cis-elements are found in a conserved order of appearance. Murozuka et al. (2018) suggested that these motifs are involved in the regulation of grain storage protein expression in barley and wheat, particularly in the binding of different transcription factors such as MYB or bZIP or positive regulators of ethylene-related signaling [11]. In our study, the motif enabling the binding of the latter (*RYREPEAT*/*EIN3* motifs) is present in multiple copies in the studied promoters. In barley, it also has been suggested that *NAC* preferentially expressed in the grain may have derived from a duplication of *NAC* genes expressed in leaves that are linked to senescence [11]. Therefore, duplicated genes may have undergone a neofunctionalization following the acquisition of specific grain-promoters. Moreover, the *GNAC* genes studied here belong to the subfamily that contains the fewest introns (NAC-d). This result suggests the existence of a similar mechanism of duplication in bread wheat. Moreover, Murozuka et al. (2018) noticed the presence of specific motifs present exclusively in “Grain NAC” in barley and absent in the senescence-associated NAC, such as the *P-BOX2* element (prolamin-box *TGTAAAG*), present in only one copy [11]. This *P-BOX* is essential for regulating the expression of seed storage proteins (SSPs) in cereals [48]. We observed this motif in the promoter of all the *Triticeae* and some *Poaceae* (*Brachypodium distachyon, Panicum hallii, Setaria italica*). Moreover, for some genes, they are present in more than one copy (e.g., the three homeologs of *TaGNAC1*). However, they have not been found in the promoters of maize, rice, sorgho or sesame *GNAC* orthologs.

During our experiments in bread wheat and einkorn, samples were taken at the early stages of grain development to study the gene’s expression during cell division (the first stage before grain filling). However, in einkorn, a later sampling point was added (700 °Cd after anthesis). At this developmental stage (belonging to the maturation–desiccation phase), the four *GNAC* gene expression decreases, suggesting that the expression of these genes does not increase exponentially during grain development but decreases during the late phases of its development. These results are in agreement with those of Murozuka et al. (2018) in barley, who observed that the expression of *GNAC* genes initiated 7–10 days after flowering as in bread wheat, with a maximum at 15–25 days after flowering, during the grain-filling period [11]. In addition, Gao et al. (2021) suggested that the early grain-filling process may be regulated by transcription factor families, such as NAC, with high expression levels 8–10 days after pollination [49]. They focus on *TaNAC019*, which showed a rapid increase in expression level from 8 days after pollination, and they demonstrate its role as a wheat grain quality regulator of starch- and seed storage protein-related genes. As *TaNAC019* is a closely related gene of *TaGNAC3* (as we can see in Figure 3, *TaNAC019* = *TraesCS3A03G0172000.1*) and they present the same expression profile, we can propose the hypothesis of a role of our *TaGNAC* in the regulation of starch and seed storage protein.

In addition, Girousse et al. (2018), who performed a WGCNA analysis, demonstrated that the four *TaGNAC* genes harbor a gene expression profile that is modified in response to heat stress and they belong to a module of expression correlated with grain length and endosperm cell number under control and heat-stressed conditions [50]. Taken together, these findings support the hypothesis that the *TaGNAC* genes may play a central role during the early phases of grain development.

### 3.3. The Four GNAC Genes Are Expressed in the Seed Earlier in Response to Thermal Stress

Globally, the four *GNAC* expression profile is altered under heat stress conditions. Interestingly, our results showed that the four *TaGNAC* transcription factors in bread wheat (*T. aestivum*) and einkorn (*T. monococcum*) present a similar expression pattern and show the strongest difference in expression level at an early developmental stage (120 °Cd in wheat, i.e., cell division stage) under heat stress condition. The expression profile of the *GNAC* genes thus appears to be conserved between these two closely related species. The 120 °Cd stage appears in the middle of the cell division phase and the impact of heat stress leads to an acceleration of the four *GNAC* expression during this early phase of grain development. These findings highlight the pleiotropic effect of four *GNAC* genes suggesting that these stress-response genes may coordinate crosstalk between environmental signals and the developmental processes to better adapt the grain compartment to the conditions. Therefore, the *GNAC* genes may be suggested as modulators of transitions in grain development that will lead to adjusting the switching time phases in response to abiotic parameters (such as heat stress) that could reduce the capacity of grain to be a sink. Furthermore, in a previous work, Guérin et al. (2019) showed that the four *TaGNAC* genes are also overexpressed in response to drought, but differently between two drought-contrasted genotypes [12].

To further investigate whether the four *TaGNAC* genes are involved in the response to heat stress, we explored the cis-motifs present in their promotors. Generally, cis-motifs of *TaNACs* known to be associated with phytohormones and abiotic/biotic stress are abundant. They include cis-motifs involved in phytohormone response (such as *CGTCA*-motif, *ERE*, *O2-site*, *TGA-element*, *GAREelement*, and *ABRE*), in biotic stress response (such as *W box*, *box S*, *RY-element*, and GCN4_motif), and in abiotic stress response (such as *MBS*, *DRE core*, *ARE*, *WUN-motif*, *LTR*, and *TC-rich* repeats) [51]. In our study, we found cis-motifs that are common to all *GNAC* from the *Triticeae* species tested here and others that are specific to one or two promoters. Among the common motifs to all *GNAC1*, *GNAC2*, *GNAC3* and *GNAC4* orthologs (=motifs 1, 4 and 7), motifs 4 and 7 carry the consensus sequence *HSFC1*. *HSFC1* is known to be involved in the heat stress response in Arabidopsis [52] but was also described as involved in the response to a wide range of abiotic stresses (heat, osmotic, salt, and cold stress in leaves and roots) in fescue for example [53]. Within motif 4, we also identified the consensus sequence *HSFB2A*, a heat shock factor involved in Arabidopsis thaliana gametophyte development [54]. A consensus promoter over the 14 *Triticeae* species is represented in Figure 8. Here, we propose a model promoter that is linked to development and stress response based on three conserved motifs (motifs 4 and 7 to stress response and motif 1 to development) identified in all 14 species tested belonging to *Poaceae*. Some authors suggested that natural variations in abiotic stress tolerance, such as drought tolerance, may be due to indel variations in the promoters of *NAC* genes (*ZmNAC111* [55]). These findings suggest a major breakthrough in determining the genetic basis underlying phenotypic variation in wheat tolerance, and the potential application of these responsive cis-element-containing indels in regulating gene expression involved in important agronomic traits. The exploration of genetic variations of the cis-element identified here could be used to test the phenotypic variations in heat tolerance.

## 4. Materials and Methods

### 4.1. Sequence Identification and Analysis in Bread Wheat

The genomic, CDS and protein sequences of the 10 available homeologs of the four bread wheat *TaGNAC* genes were retrieved from the RefSeq v2.1 database of the International Wheat Genome Sequencing Consortium [56]. The three homeologs of *TaGNAC1*, *TaGNAC2* and *TaGNAC4* were identified. For *TaGNAC3*, only the copy carried by chromosome 5A has a Traes id on v2.1 of the bread wheat pseudomolecule (Appendix A). Chromosomal anchorage of the 10 *TaGNAC* homeologs was performed using the MapChart software [57], and chromosomal length and gene location provided by the IWGSC.

We used the RefSeq v2.1 protein database to retrieve the 488 Traes id identified in bread wheat (in the RefSeqv1.0 database) by Guérin et al. (2019) [12]. Only 454 sequences were retrieved, the others probably have been removed during the database updates. An unrooted phylogenetic tree of those 454 *TaGNAC* sequences was designed using the maximum-likelihood method available in MEGA-X software [58], based on the alignment of the protein sequences using the ClustaW method. Then, the sequences anchored in the clade containing the homeologs of our four *TaGNACs* were retrieved. On the MEME Suite platform (http://meme-suite.org/tools/meme, accessed on 15 January 2022), we used the MEME tool to search for the presence of conserved motifs among these protein sequences with parameters as follows: motif number = 20, width = 10–150 amino acids. Subsequently, we imported these motifs into the FIMO tool to check for their presence in the 64 orthologous sequences of bread wheat *TaGNACs* (the results were filtered using a *p*-value < 1 × 10^−10^).

### 4.2. Sequence Identification in Einkorn from Bread Wheat

Unlike bread wheat, there is no recent annotation of einkorn (the most recent was in 2014 [59]). In order to clone orthologous sequences to *TaGNAC1*, *TaGNAC2*, *TaGNAC3* and *TaGNAC4* in einkorn, we used primers drawn on the A homeolog of each of the four *TaGNACs* to amplify their einkorn orthologs (genotype TM35821) by PCR, using the Amplitaq enzyme and according to the manufacturer’s instructions. Four sequences were amplified, sequenced, and named *TmGNAC1*, *TmGNAC2*, *TmGNAC3* and *TmGNAC4* for *Triticum monococcum GNAC*. The tertiary structure of *GNAC* genes from bread wheat and einkorn were modelled using the RaptorX tool (http://raptorx.uchicago.edu/StructPredV2/predict/, accessed on 25 January 2022).

### 4.3. Identification of the Four TaGNAC Orthologs in the Genome of 13 Other Cereals

The proteomes of 13 cereal species were retrieved from the EnsemblPlant website (http://plants.ensembl.org, accessed on 5 February 2022). Then, by performing a blastP of the three homeologs of the four GNAC proteins against the whole proteome of each species and selecting the best hit for each result, we identified their orthologous sequences in the 13 sequences (Appendix A). The cDNA sequences and the 2000 bp upstream of the cDNA were recovered from EnsemblPlant, for each of the 10 bread wheat *TaGNACs* genes and their orthologous sequences in the 13 cereal species. The bread wheat promoter sequences (2000 bp) were retrieved in the RefSeq v1.1. Due to the absence of an annotated genome in einkorn, *Triticum monococcum*, we were unable to retrieve their promoter sequence.

We searched for conserved motifs in promoter sequences, using MEME Suite (https://meme-suite.org, accessed on 26 February 2022) with parameters as follows: motif number = 10, width = 10–150 bp. On these motifs, the TOMTOM analysis of MEME Suite was performed using the default parameters [60], the PBM motifs database from *Arabidopsis thaliana* [61] and an e-value < 10. An unrooted phylogenetic tree was designed using the maximum-likelihood method available in MEGA-X software, based on the alignment of the GNAC protein sequences using the ClustalW method. The presence of the NAC domain (IPR003441, PS51005 and PF02365) on the protein sequences of TaGNAC, TmGNAC and 13 other cereal species was verified by InterProScan and the presence of intracellular localization signals was studied using the Phobius software.

### 4.4. Plant Material and Thermal Stress Treatment

In order to study the *TaGNAC* expression profile in optimal growth and under thermal stress, we used the RNA samples generated in an experiment described in Girousse et al. (2018) [50]. RNA samples were analyzed at six developmental stages (0, 80, 120, 180, 220 and 300 °Cd after anthesis) in the control conditions (19 °C) and in response to a moderate thermal stress (27 °C) applied during 10 days after anthesis.

To compare the *GNAC* expression between bread wheat and einkorn, an einkorn culture with seeds of the TM35821 genotype was performed in a greenhouse using the same cultural conditions as Girousse et al. (2018) in wheat [50]. Spikes were harvested at eight stages (0, 40, 80, 120, 180, 220, 300 and 700 °Cd after anthesis) and frozen into liquid nitrogen. Total RNA were isolated and purified from grains, according to the protocol described by Capron et al. (2012) [62].

### 4.5. RNA Extraction and RT-qPCR of Candidate Genes

For bread wheat and einkorn RNA samples, reverse Transcription was performed on 2.5 μg of total RNA, using the Thermo Scientific Maxima First Strand cDNA Synthesis Kit, and according to the manufacturer’s instructions, then an RNAse H (Thermo Scientific 18021-071, Waltham, Massachusetts, USA) treatment was performed. The expression of *GNAC* genes was realized using real-time quantitative polymerase chain reaction (RT-qPCR) using specific primers designed on the coding sequences (listed in Appendix A), with PrimerQuest (https://eu.idtdna.com/, accessed on 15 January 2022). RT-qPCR were carried out in a 15 μL volume containing 25 ng of cDNA, 7.5 nM primers and 7.5 μL of LightCycler 480 SYBR Green I Master (Roche Diagnostics #04887352001). The thermocycler LightCycler 480 system (Roche, Bâle, Switzerland) of the Gentyane platform, INRA de Crouël, France (http://gentyane.clermont.inra.fr/, accessed on 3 March 2022) was used. The thermal cycle was set up as follows: pre-incubation at 95 °C/10 min, 40 amplification cycles of 95 °C/10 s, 15 s of primer optimal temperature, then 72 °C/15 s. Relative gene expression was determined by the 2^-ΔΔCt^ method [63], using an RNase L inhibitor-like protein (*Ta2776*) for normalization [64] in bread wheat. Unlike bread wheat, no extensive knowledge is known in einkorn about housekeeping genes. Thus, we use the primers designed by Paolacci (2009) for three high-potential housekeeping genes from bread wheat (*Ta2776*, *Ta2291* and *Ta54227*) on the einkorn samples [64]. Among the three potential genes, *Ta2776* was selected as the best housekeeping gene. To estimate the stress response intensity, the ratio expression in the stressed condition/expression in the control condition was calculated for each homeolog and each studied stage.

## 5. Conclusions

Our study aimed at characterizing four *NAC* genes that are preferentially expressed in the grain (called *GNAC* for “Grain-NAC”). A high conservation of their structure was identified, in terms of genomic and protein organization, through 15 species of cereals belonging to the *Triticeae* tribe, and this conservation may be specific to this tribe. In bread wheat and einkorn, the expression profile during seed development is similar for the four *GNAC* genes. This profile is accelerated in response to heat stress, and that may affect the seed development. We then hypothesized that the four *TaGNAC* may play an important role in the accumulation of storage molecules (most probably storage proteins) in the grain of bread wheat as integrators of environmental stress signals, and this pattern may be also conserved through the *Triticeae* tribe. Their role may potentially be linked to the fine tuning of the regulation between heat and drought stress responses and the most efficient grain development, particularly during grain filling. Singh et al. (2021) demonstrated that the understanding of the molecular mechanisms of a plant’s development and its response to stress, under the control of *NAC* genes, is very promising for improving the tolerance of future food crop plants to abiotic stresses [1]. The fact that they are preferentially expressed in the grain compartment suggests that they might be good candidates for the improvement of grain yield in wheat in genetic engineering.

## Figures and Tables

**Figure 1 ijms-23-11672-f001:**
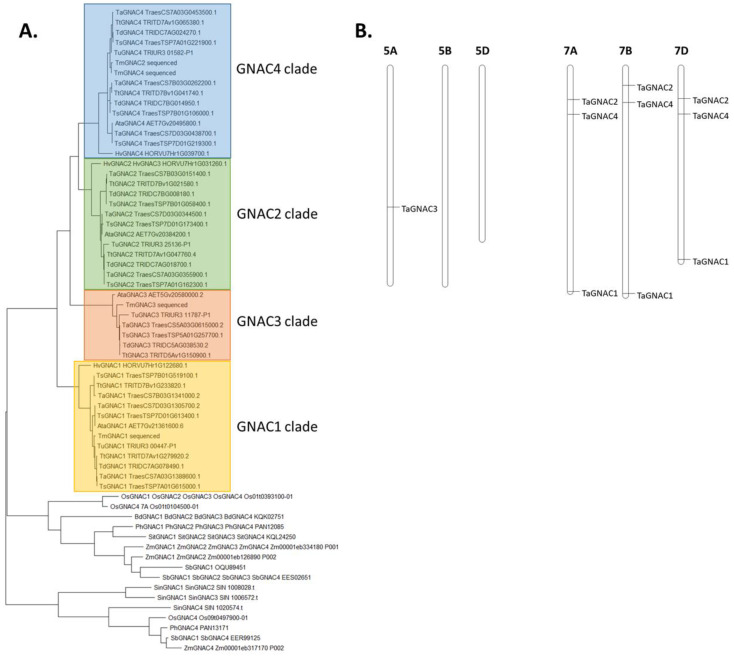
(**A**) Unrooted phylogenetic tree of the four TaGNAC of bread wheat (i.e., 10 sequences) and their orthologous sequences amplified in *T. monococcum* and identified by blast in 13 other cereals. Ata: *Aegilops tauschii*, Bd: *Brachypodium distachyon*, Hv: *Hordeum vulgare*, Os: *Oryza sativa*, Ph: *Panicum hallii*, Sb: *Sorghum bicolor*, Sin: *Sesamum indicum*, Sit: *Setaria italica*, Ta: *Triticum aestivum*, Td: *Triticum dicoccoides*, Tm: *Triticum monococcum*, Ts: *Triticum spelta*, Tt: *Triticum turgidum*, Tu: *Triticum urartu*, Zm: *Zea mays*. (**B**) Chromosomal anchorage of the three homeologs of *TaGNAC1*, *TaGNAC2* and *TaGNAC4* on chromosome 7 and the only-known homeolog of *TaGNAC3* on chromosome 5.

**Figure 2 ijms-23-11672-f002:**
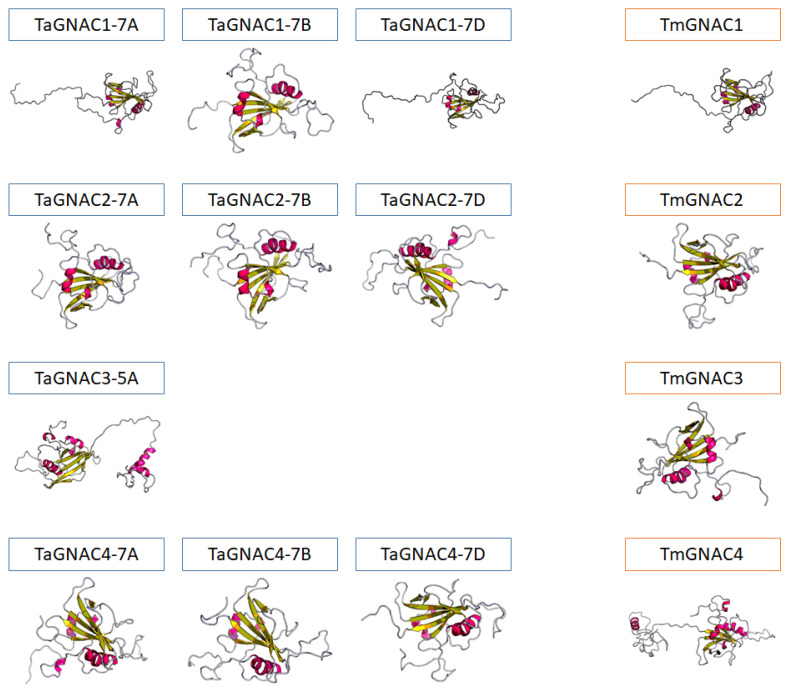
Tertiary structure of the TaGNAC and TmGNAC proteins. Pink: alpha-helix; yellow: beta-sheets.

**Figure 3 ijms-23-11672-f003:**
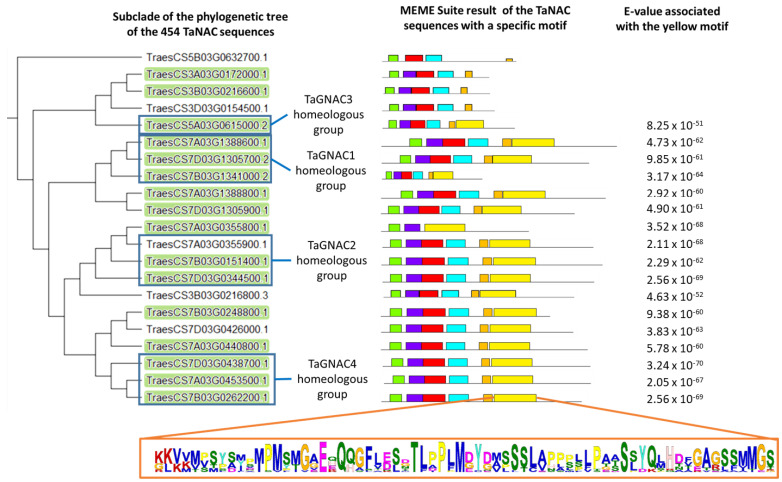
Clade extracted from the phylogenetic tree carrying the 454 high confidence sequences of the TaNAC family of bread wheat, identified in the new RefSeq v2.1 database. The homeologous sequences of the *TaGNAC1*, *TaGNAC2*, *TaGNAC3* and *TaGNAC4* genes existing in this database are framed in blue. Of the 19 sequences expressed preferentially in the grain, 16 are carried by this clade (highlighted in green). Within this clade, 17 sequences carry a specific conserved pattern (yellow block), presented in Logo format at the bottom of the figure.

**Figure 4 ijms-23-11672-f004:**
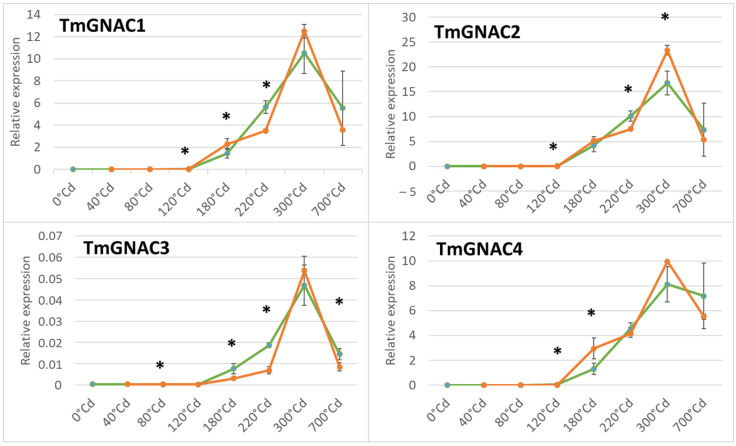
Expression profile of the *TmGNAC1*, *TmGNAC2*, *TmGNAC3*, and *TmGNAC4* genes in einkorn grain during eight developmental stages. The expression was monitored in control (19 °C, green curve) and moderate thermal stressed condition (27 °C during 10 days after anthesis, orange curve). * = significant difference between control and stressed conditions (*T* test, *p*-value < 0.05). The significant difference observed at 120 °Cd after anthesis for the *TmGNAC1*, *TmGNAC2* and *TmGNAC4* reveals a higher expression in the control condition compared to the stressed condition.

**Figure 5 ijms-23-11672-f005:**
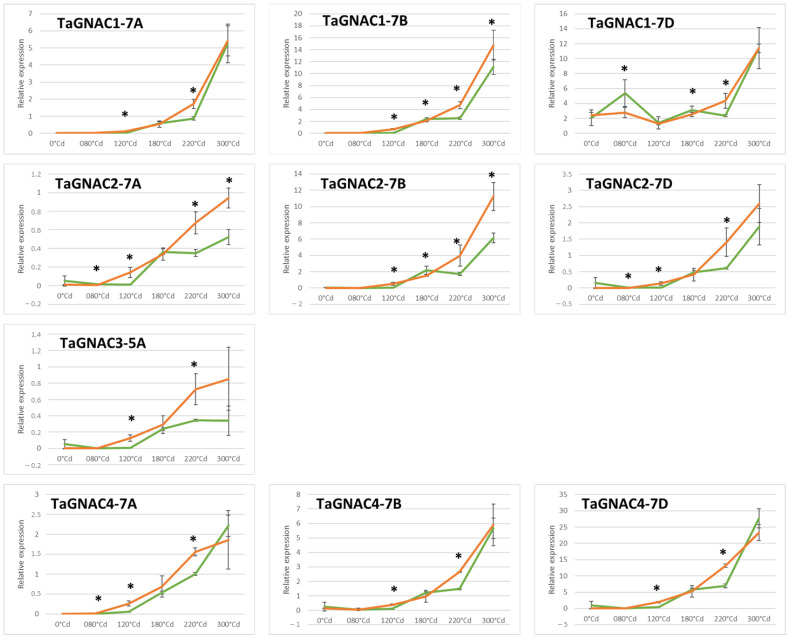
Expression profile of the homeologs of *TaGNAC1*, *TaGNAC2*, *TaGNAC3*, and *TaGNAC4* in bread wheat grain during six developmental stages. The expression was monitored in control (19 °C, green curve) and moderate thermal stressed condition (27 °C during 10 days after anthesis, orange curve). * = significative difference between control and stressed conditions (*T* test, *p*-value < 0.05).

**Figure 6 ijms-23-11672-f006:**
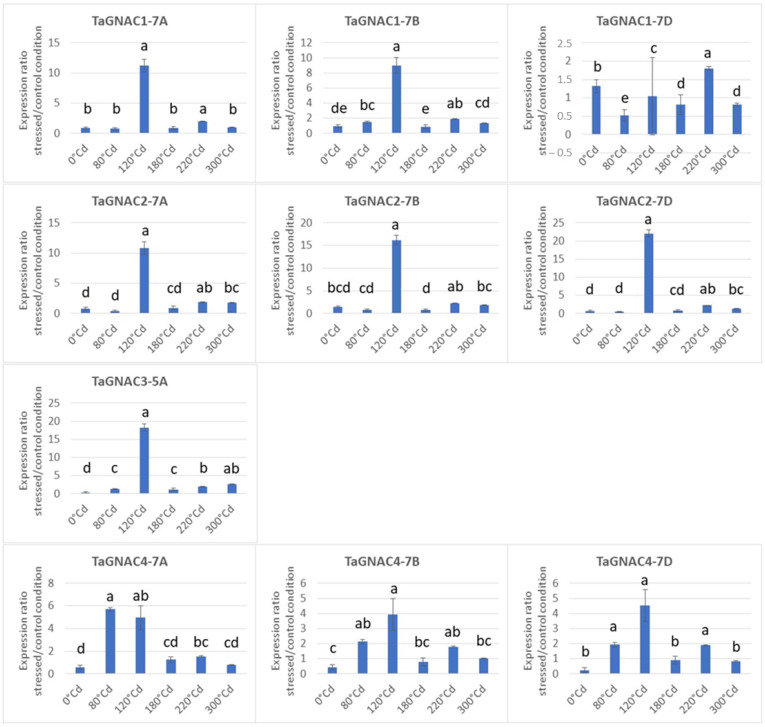
Ratio of the three homeologs of *TaGNAC1*, *TaGNAC2*, *TaGNAC3* and *TaGNAC4* expression in response to moderate heat stress vs. their expression in control condition. Statistical test = Kruskal Wallis, *p* < 0.05. The same letter means there is no significant difference.

**Figure 7 ijms-23-11672-f007:**
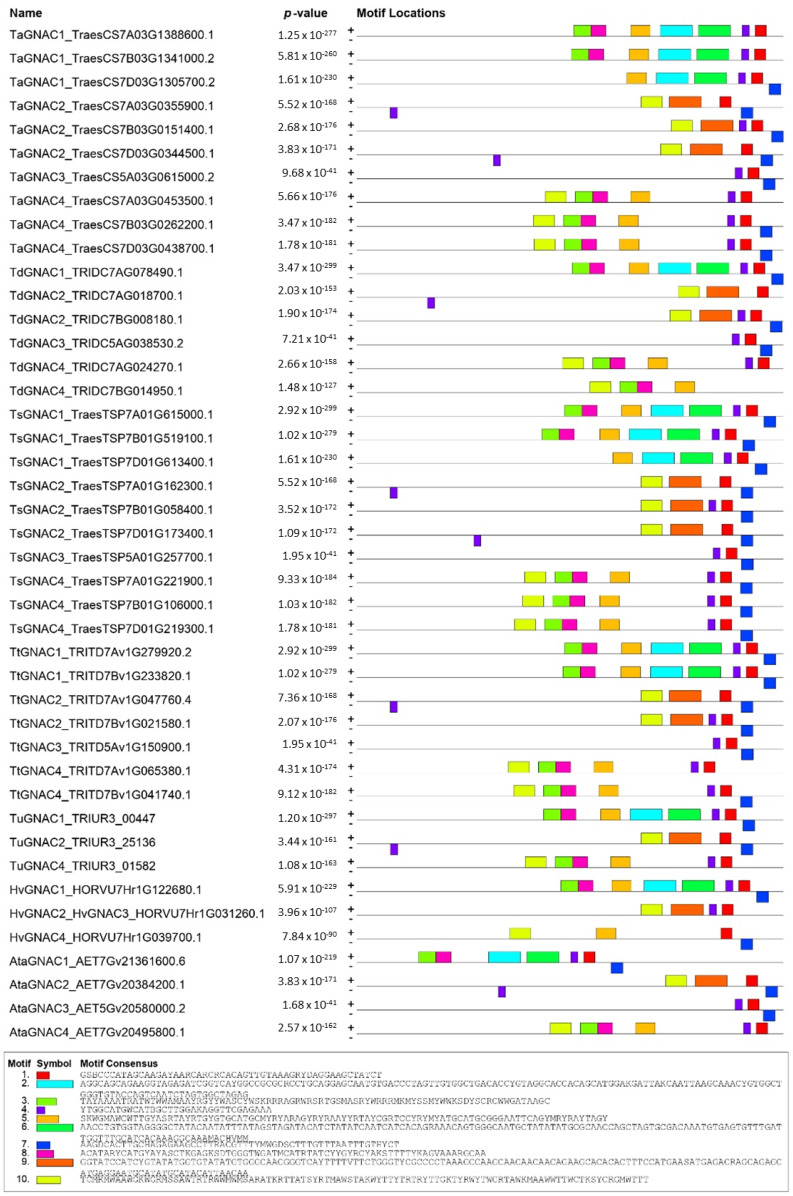
Identification of conserved motifs in the promoter (2000 bp upstream of the CDS) of the four bread wheat *TaGNAC* genes homeologs and their orthologs in 13 cereal species. Ata: *Aegilops tauschii*, Bd: *Brachypodium distachyon*, Hv: *Hordeum vulgare*, Os: *Oryza sativa*, Ph: *Panicum hallii*, Sb: *Sorghum bicolor*, Sin: *Sesamum indicum*, Sit: *Setaria italica*, Ta: *Triticum aestivum*, Td: *Triticum dicoccoides*, Ts: *Triticum spelta*, Tt: *Triticum turgidum*, Tu: *Triticum urartu*, Zm: *Zea maya*.

**Figure 8 ijms-23-11672-f008:**
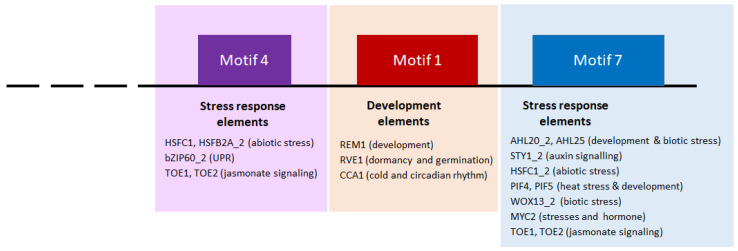
Schematic representation of the cis-elements belonging to the three identified conserved motifs among the 14 orthologous *Triticeae GNAC* genes. Only the 3′ terminal part of the promoter is represented (stopped at the beginning of the 5′UTR). The motif 1 gathers cis-elements that may be involved in plant development. Motifs 4 and 7 present cis-elements that are involved in stress responses.

## Data Availability

Not applicable.

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
