# Peer review of "Insights into Four NAC Transcription Factors Involved in Grain Development and in Response to Moderate Heat in the Triticeae Tribe"

_ijms, 2022, doi:10.3390/ijms231911672_

Round 1

Reviewer 1 Report

The authors of this study investigated four genes belonging to NAC family in bread wheat which are specifically expressed in the grains during the reproductive stage. They characterized these genes through various molecular and bioinformatics methodologies and concluded that these genes may be linked to fine tuning of the regulation between heat and drought stress responses and the grain development during grain filling. The manuscript is well written with apt methodologies and following results and their discussions. I do not have any major concerns about the manuscript.

Reviewer 2 Report

Dear Authors,

Please find the comments below for your reference.

Title: Molecular characterization of four genes encoding NAC transcription factors involved in the grain development and its response to moderate heat stress in the Triticeae tribe

The manuscript provides insight into the role of the NAC transcription factor in grain development and heat stress. However, the overall manuscript needs major revision in writing. The manuscript has enough data, but it’s not explained properly. The information is scattered and lacks proper flow in writing. It is very hard to follow. Authors have repeated methods and literature in each section, including results. The manuscript, therefore, looks very lengthy. Here are some specific comments:

Title: The title is too long. Authors are suggested to put a concise title. 

Abstract:

Abstract lacks scientific writing. Please modify the sentences starting from ‘We’. The abstract needs to be reframed. Please include why only four NAC were chosen. What is the problem statement, hypothesis, and objectives?

Line 13-14: ‘We then focused on the expression of TaGNAC and einkorn TmGNAC under control conditions (19°C) and in response to thermal stress (27°C) during grain’. What were the conditions? Was it only thermal conditions? Please reframe the statement.

Keywords: Please arrange the keywords in alphabetical order.

Introduction: Overall, the introduction is lengthy and contains a lot of information which are not described properly. Authors have to re-think and focus on what the manuscript is about. The title states that ‘grain development and heat stress, the problem statement and objectives should be focused on these two topics rather than discussing all kinds of biotic and abiotic stress. Sentences are framed as if written for ‘Review of Literature. There are random sentences without references such as:

Line 18: Reference is missing for the statement ‘NAC (NAM (No Apical Meristem) – ATAF (Arabidopsis Transcription Activation Factor) – CUC (Cup-shaped Cotyledons)) are one of the largest family of transcription factors in the plant kingdom’

Line 28-31: Authors have described NAC TFs as one of the largest families and used different terminology to denote the presence in plants. The family is followed by members or genes. Please edit to make the information consistent.

Methods:

Section: Plant material and thermal stress treatment

Please state: Seeds or clonally propagated plants were used, germination and growth conditions, conditions of growth, heat stress treatment and how many replicates were analyzed.

Result:

This section has enough details but contains methods in each paragraph. Please edit this section keeping only results and remove all the repeated information from the methods.

Figure labels: If the authors want to put two figures under the same labels, please label them as ‘(a)’ and ‘(b)’ and not ‘left’ and ‘right.’ Also, please remove the method part from the label; just explain what is seen in the figure. Please check the labels of all the figures and remove the method part from each label.

Gene names: Please check the gene names and modify them as italics.

Discussion:

This section also has a lot of literature reviews. Authors are suggested to discuss only the results. Please state your result first and then compare it with published information, rather than writing the available information and supporting the statement.

For example, this section (Line 458- 465):

‘Evolutionary patterns of gene families vary between plant taxa. Jin et al. (2017b) observed two distinct evolutionary scenarios of NAC transcription factors between dicots and grasses [46]. According to these authors, dicotyledonous lines show an acquisition of a larger number of genes than grass lines, while the latter shows less genetic loss. Within grasses, the expansion of transcription factor families varies between species. We have shown that several GNAC sequences in bread wheat are orthologs of one gene in rice, maize, or Brachypodium distachyon for example. This result is consistent with the observation of Murozuka et al. (2018), who unraveled the expansion of a particular clade within the Triticeae compared to other Poaceae species (rice and maize) [10]’.

The authors should state their findings first, and then support the details with available literature. Unlike this paragraph, which starts with published information and then supports the information with the result obtained. This should be followed in the overall writing of the discussion.

Conclusion:

Why this section contains a reference in Line 751? Authors are suggested to remove all “We” and any references from this section.

Line 743 – 745: ‘We identified a clear expression profile that may be correlated to the grain filling stage, and that is conserved, but accelerated in response to heat stress’. This statement is misleading, as the authors should clarify why they have used the term ‘may.’ Also, what are the other parameters that need to be studied to confirm the role of GNAC to be used as a candidate for the improvement in the grain yield? What is the prospects of this study?
